# Patterning Configuration of Surface Hydrophilicity by Graphene Nanosheet towards the Inhibition of Ice Nucleation and Growth

**Biao Jiang [1], Yizhou Shen [1,\*], Jie Tao [1,\*], Yangjiangshan Xu [1], Haifeng Chen [2], Senyun Liu [3], Weilan Liu [4] and Xinyu Xie [1]**

[1]   College of Materials Science and Technology, Nanjing University of Aeronautics and Astronautics, 29 Yudao Steet, Nanjing 210016, China; jiangbiao@nuaa.edu.cn (B.J.); xyjshan@nuaa.edu.cn (Y.X.); xiexinyu@nuaa.edu.cn (X.X.)

[2]   Department of Materials Chemistry, Qiuzhen School, Huzhou University, 759# East 2nd Road, Huzhou 313000, China; headder@zjhu.edu.cn

[3]   Key Laboratory of Icing and Anti/De-Icing, China Aerodynamics Research and Development Center, Mianyang 621000, China; liusenyun@cardc.cn

[4]   Institute of Advanced Materials, Nanjing Tech University, 30 Puzhu South Road, Nanjing 210009, China; iamwlliu@njtech.edu.cn

\*   Correspondence: shenyizhou@nuaa.edu.cn (Y.S.); taojie@nuaa.edu.cn (J.T.)

**Abstract:** Freezing of liquid water occurs in many natural phenomena and affects countless human activities. The freezing process mainly involves ice nucleation and continuous growth, which are determined by the energy and structure fluctuation in supercooled water. Herein, considering the surface hydrophilicity and crystal structure differences between metal and graphene, we proposed a kind of surface configuration design, which was realized by graphene nanosheets being alternately anchored on a metal substrate. Ice nucleation and growth were investigated by molecular dynamics simulations. The surface configuration could induce ice nucleation to occur preferentially on the metal substrate where the surface hydrophilicity was higher than the lateral graphene nanosheet. However, ice nucleation could be delayed to a certain extent under the hindering effect of the interfacial water layer formed by the high surface hydrophilicity of the metal substrate. Furthermore, the graphene nanosheets restricted lateral expansion of the ice nucleus at the clearance, leading to the formation of a curved surface of the ice nucleus as it grew. As a result, ice growth was suppressed effectively due to the Gibbs–Thomson effect, and the growth rate decreased by 71.08% compared to the pure metal surface. Meanwhile, boundary misorientation between ice crystals was an important issue, which also prejudiced the growth of the ice crystal. The present results reveal the microscopic details of ice nucleation and growth inhibition of the special surface configuration and provide guidelines for the rational design of an anti-icing surface.

**Keywords:** surface hydrophilicity; graphene; ice formation; clearance; molecular dynamic simulation

## 1. Introduction

Understanding and regulating the crystallization of supercooled water on surfaces is essential in both basic research and engineering applications [1,2]. The freezing of water on surfaces is, in fact, a complicated phenomenon that requires collective understanding of nucleation, crystal growth, surface science and thermodynamics [3–5], and plenty of research has been done theoretically and experimentally [6,7]. According to previous studies, ice nucleation can be affected by many factors, including surface morphology [8,9], wettability [10,11], shear flow [12], ions and contamination particles [13–15], etc. For example, Yue et al. and Wang et al. found that micro-hierarchical structures or patterns on a silicon surface would delay ice nucleation due to enhanced free energy for nucleation [16,17]. He

et al. reported the effect of counter ions on heterogeneous ice nucleation on a polyelectrolyte brush and discovered that a distinct efficiency of ions in turning ice a freezing temperature follows a certain sequence [18].Their study revealed that counter ions have a profound ion-specific effect on the relaxation of the hydrogen bond and the formation rate of interfacial water molecules.

Over the past few years, it has been investigated that graphene and its derivatives could influence ice nucleation and growth [19–21]. The two-dimensional material, graphene, has become one of the most promising materials in recent decades due to its potential applications in high performance electronics, sensors and energy storage devices [22–27]. It can be made into fibers, membranes and can be drop-casted onto various substrates [28–30]. Using a molecular dynamics simulation, Lupi and Molinero studied the heterogeneous ice nucleation of liquid water in contact with graphitic surfaces of various dimensions and curvatures, which were reported in experimental characterizations of soot [31,32]. Their results indicated that the ordering of interfacial liquid water on a graphene surface was the main reason for the facilitated heterogeneous nucleation [33]. Not only a promoting material, graphene and its derivatives can also restrain ice formation under specific conditions [22,34]. The exceptional Joule's heating effect and electrothermal effect of graphene-based composites have received much attention in the design of anti-icing systems [35,36]. The lightweight graphene-based material is expected to be an ideal heater to prevent ice freezing. Specifically, there are numerous studies taking advantage of these properties for anti-icing and de-icing applications [37,38].

Recently, graphene oxide (GO) was used to mimic antifreeze proteins (AFPs) and considered to be an advanced icing inhibitor [39–42]. The repeated hexagonal carbon ring structure arranges the functional groups on the basal plane of GO to match with an ice crystal lattice, leading to the preferred adsorption of GO on existing ice in liquid water. Thus, the growth of the ice crystal is suppressed owing to the Gibbs–Thomson effect; that is, the curved surface lowers the freezing temperature [43,44]. The curved surface of the ice crystal derives from the special surface configuration, which involves different surface hydrophilicity and a crystal structure between the two materials. By using molecular dynamics simulations, Zhang and Chen studied ice nucleation on graphene surfaces functionalized by several kind of ions and methane molecules [45]. Their results indicated that the ice nucleation ability of the functionalized surfaces was weakened compared with that of the smooth graphene surface, depending on the type and the number of functional groups. Akhtar et al. presented an anti-icing coating based on fluorinated graphene, which could strikingly delay ice formation in a high humidity environment [46]. The anti-icing performance of fluorinated graphene was attributed to a robust liquid layer arising from the interface confinement effect that increases the ice-water contact angle and viscosity of water molecules near the surface. Additionally, the coupling of surface crystallinity and surface hydrophilicity was found to be a controlling factor for heterogeneous ice nucleation [7]. With an appropriate hydrophilicity, the arrangement of the water layer in contact with crystalline graphene can be changed; thus, the ice nucleation rate decreases consequently. The interplay between surface morphology and hydrophobicity on heterogeneous ice nucleation was also studied by Martin [4]. They showed that lattice mismatch of the surface with respect to ice is desirable for a good ice nucleating agent. Hence, it is interesting to investigate the synergistic effect of a surface hydrophilicity discrepancy and the clearance configuration created by an anchored graphene sheet on ice nucleation and growth.

In this work, a graphene nanosheet was introduced to design a special surface configuration of surface hydrophilicity on a metal surface. The characteristics of ice nucleation and growth on the surface were investigated using a molecular dynamics (MD) simulation. We constructed a series of surface configurations with various surface hydrophilicity discrepancies between a graphene nanosheet and metal substrate and explored the initiation of ice nucleation and growth processes under specific surface conditions. Besides, we studied the growth process of an ice nucleus and computed the growth rate of the ice nucleus. A

different surface configuration resulted in varying levels of ice growth inhibition. We also discussed the misorientation between grain boundaries of the ice crystal [47].

## 2. System and Simulations

The simulation system contained four graphene nanosheets anchored on a metal substrate, which were cleaved in the (100) surface of a face-centred cubic (fcc) crystal. The lattice parameter $a_{fcc}$ was 4.0495 Å, and 6189 water molecules were covered on this surface, as shown in Figure 1a. The simulation box had dimensions of $L_x$ = 9.758 nm, $L_y$ = 9.744 nm and $L_z$ = 10.000 nm. A void space was set in the simulation box to avoid the influence of a periodic boundary condition in the z direction. Four graphene nanosheets with a size of 2.85 nm × 3.27 nm were fixed atop of the metal substrate with a distance of 6 Å [33,48], resulting in a 2-nm-wide cross-shaped clearance between the graphene nanosheets, as shown in Figure S1.

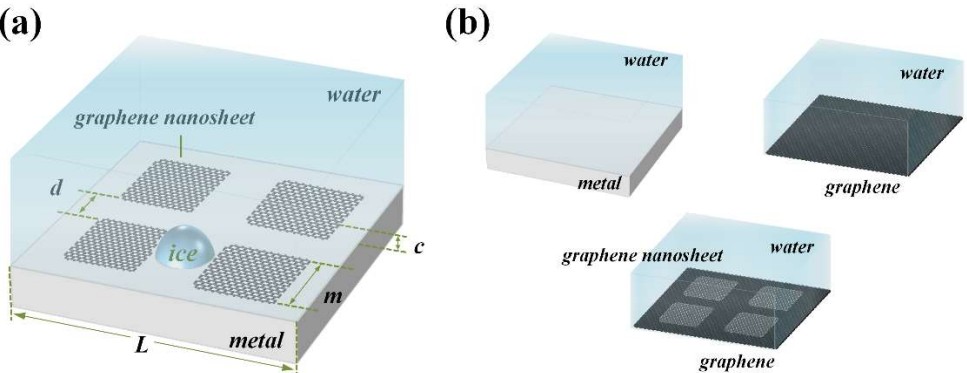

**Figure 1.** Illustrations of simulation systems. (**a**) Model of the metal–graphene nanosheet surface that was covered with a box of liquid water. A typical ice nucleus formation is illustrated on the surface. (**b**) Models of three other simulation systems. Pure metal, pure graphene and graphene–graphene nanosheet surface systems.

The system described above is defined as a metal–graphene nanosheet system. Additionally, we constructed three other simulation systems for comparison, which were pure metal, pure graphene and graphene–graphene nanosheet surface systems. Details are shown in Figure 1b and Table S1.

The coarse-grained monatomic water (mW) model was employed in this paper to describe the interaction between water molecules [49]. This specific water model has excellent structural properties and a melting point close to the experiment. The mW model treats water molecules as a single particle that interacts through short-range two-body and three-body interactions [50]. Since it is monatomic, it exhibits faster dynamics, which allows for the alleviation of computational costs of our simulations for the ice freezing process. Detailed information about the mW water model is provided in Supplementary Note 1. The Lennard-Jones (LJ) potential was used to model the interaction between water molecules and substrate atoms. Length and energy parameters, $\sigma_{w\text{-}g}$ and $\varepsilon_{w\text{-}g}$, were set to 2.488 Å and 0.13 kcal/mol, respectively, for each water molecule interacting with carbon atoms [31]. Otherwise, for interactions between water molecules and metal atoms, $\varepsilon_{w\text{-}m}$ was tuned from 0.13 kcal/mol to 14.0 kcal/mol in each simulation to obtain a different surface hydrophilicity of the metal substrate, while $\sigma_{w\text{-}m}$ was fixed in 2.8798 Å [51]. As a result, the difference between $\varepsilon_{w\text{-}g}$ and $\varepsilon_{w\text{-}m}$ is defined as surface hydrophilicity discrepancy $D_\varepsilon$. There is no need to define the metal–metal and carbon–carbon interaction potentials, because they are fixed in the simulation system. Periodic boundary conditions were applied in three dimensions, and the integration time-step for the velocity Verlet algorithm was set to 5 fs. After 0.2 ps of relaxation at temperature 290 K, the whole system was cooled down to 200 K gradually with a cooling rate of 0.9 K/ns in the canonical ensemble (NVT), and the

process of water freezing was studied during the quenching period. To obtain statistically reasonable results, we performed 5 repetitions of each cooling simulation for every system. All MD simulations were performed using the LAMMPS simulation package. Phase and structure identification was carried out by the Identify Diamond Structure modifier in OVITO software (version 3.6.0) during the freezing process of liquid water on each surface configuration [52]. The initiation and ending times of the icing process were recorded through the entire simulation. The initiation time of icing was identified as the moment when the number of water molecules in the ice nucleus ($N_{ice}$) started to increase rapidly. Conversely, the ending time of icing was identified as the moment when $N_{ice}$ tended to be stable, which is marked in Figure 2a.

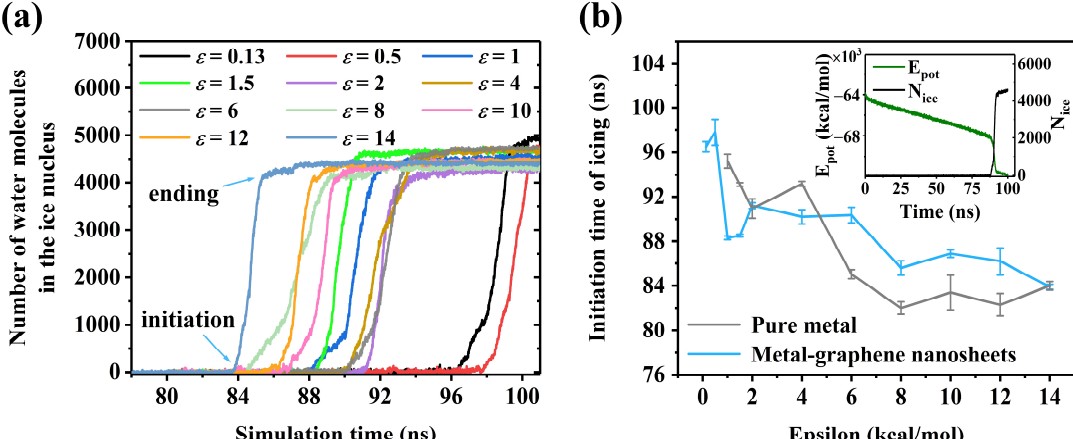

**Figure 2.** Freezing of liquid water in different surface conditions. (**a**) Number of water molecules in the ice cluster on metal–graphene nanosheet surfaces with different surface hydrophilicities. The interaction strength between the metal and water molecule $\varepsilon_{w-m}$ varied from 0.13 to 14.0 kcal/mol. (**b**) Initiation time of icing with different surface hydrophilicities of a metal substrate. Inset shows total potential energy $E_{pot}$ of the system and the number of water molecules in the ice nucleus, $N_{ice}$, during the whole simulation time. The data refer to the metal–graphene nanosheet system, and $\varepsilon_{w-m}$ was fixed to 1.0 kcal/mol in the inset of (**b**).

## 3. Results and Discussion

### 3.1. Surface Hydrophilicity and Ice Nucleation

The role of surface hydrophilicity [53,54] has been a central argument to determine whether ice nucleation and the subsequent growth process can happen [55–57]. In order to understand the impact of surface clearance on ice formation where hydrophilicity is different from peripheral graphene sheets, we performed a series of MD simulations using the interaction strength $\varepsilon_{w-m}$ as the collective variate to explore a desirable freezing delay surface, as shown in Figure 1a.

Ice formation can be obviously observed on every metal–graphene nanosheet surface as the system cooled down gradually, and the details are shown in Figure S3. The freezing process of liquid water on each metal–graphene nanosheet surface configuration is shown in Figure 2a. Curves with different colors represent various surface hydrophilicities of the metal substrate $\varepsilon_{w-m}$. Generally, lower interaction strength between a solid atom and water molecule results in later ice nucleation, which has been demonstrated in other research [54]. However, there was no simple trend for the triggering time of ice nucleation for $\varepsilon_{w-m}$ ranging from 1.0 to 6.0 kcal/mol, as shown in Figure 2b. Water molecules in the simulation system could interact with substrate atoms freely when $\varepsilon_{w-m}$ was smaller than 1.0 kcal/mol, as shown in Figure 3a. Note that a layer of water molecules was arranged in an fcc structure gradually with the increasing $\varepsilon_{w-m}$, as shown in Figure 3b. This interfacial water layer was formed adjacent to the substrate with a distance of about 1.88 Å, and it could be detected by the count of water molecules along the z direction, as shown in Figure 3b.

On one hand, water molecules in this layer were closely packed in an fcc structure that differed from the structure of an ice crystal, which is not beneficial for ice nucleation. On the other hand, other free water molecules were separated from the metal substrate by the interfacial water layer, which decreased the interaction between free water molecules and the metal substrate. Therefore, the interfacial water layer acts as a barrier and hinders the nucleation of an ice nucleus [7,58]. Meanwhile, the surrounding conditions of the graphene nanosheets were not beneficial to ice nucleation, owing to the extremely low surface hydrophilicity of 0.13 kcal/mol. Therefore, ice nucleation was delayed for $\varepsilon_{w-m}$ ranging from 1.0 to 6.0 kcal/mol because of the hindering effect of the interfacial water layer formed on the metal substrate.

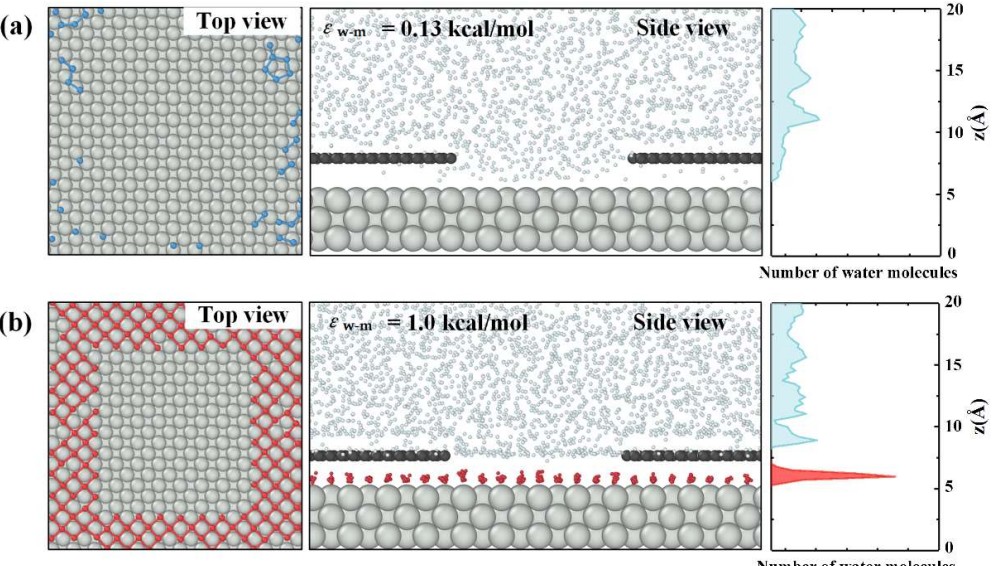

**Figure 3.** The formation of an interfacial water layer on a metal–graphene nanosheet surface. (**a**) Snapshot of the interface between a metal substrate and liquid water. The dense interfacial water layer was not formed at $\varepsilon_{w-m}$ = 0.13 kcal/mol. (**b**) Snapshot of the interfacial water layer viewed from the top and the side. The right panel is the number of water molecules along the z direction before water freezing. In all cases, the interfacial water layer is colored red for ease of visualization.

However, with the $\varepsilon_{w-m}$ increasement, a higher surface hydrophilicity of a metal substrate forced more water molecules to arrange in an orderly manner, which was convenient for the formation of an ice structure during the $\varepsilon_{w-m}$ value range from 6.0 to 14.0 kcal/mol. The competition of both effects lead to the nonlinear dependence, as shown in Figure 2b. Therefore, a proper surface hydrophilicity discrepancy between a graphene nanosheet and metal substrate is considered as a desirable agent for the delayed initiation of ice nucleation. The inset shows the total potential energy of a system changed with simulation time, together with the number of water molecules in the ice nucleus $N_{ice}$ [59]. As a plunge of the potential energy $E_{pot}$ of the system occurred, $N_{ice}$ increased suddenly, indicating the nucleation of an ice embryo. When $\varepsilon_{w-m}$ was set to 6.0 kcal/mol, i.e., a discrepancy of the surface hydrophilicity $D_\varepsilon$ of 5.87 kcal/mol, the ice nucleation on the metal–graphene nanosheet surface was delayed by 5.25 ns, which was a maximum value compared to that on a pure metal surface, as shown in Figure 2b. We noticed that a higher extent of hydrophilicity could induce a greater thickness of the interfacial water layer. The hindering effect of the interfacial water layer prevailed again, which well explained the jump of the triggering time of ice nucleation for $\varepsilon_{w-m}$ values between 6.0 to 8.0 kcal/mol.

It should be mentioned that small ice nuclei form and expand constantly under the action of structure and energy undulations in supercooled water [60]. Additionally, these unstable ice nuclei prefer to form at the edge of graphene sheets, where the energy barrier

for ice nucleation is lower compared with those in other positions [19,61]. The edge of the graphene sheet is a phase contact area where water molecules interact with carbon and metal atoms simultaneously.

### 3.2. Priority of Ice Nucleation

We noticed that the position of initial ice nucleation on the metal–graphene nanosheet surface depended on the surface hydrophilicity of the metal substrate. For $\varepsilon_{w\text{-}m}$ lower than 1.0 kcal/mol, ice nucleation tended to happen at the top of the graphene nanosheets, as shown in Figure 4e and Figure S4. On this condition, ice nucleation was triggered by the repeated hexagonal carbon ring structure of the graphene nanosheet, which matched better with an ice structure compared to the fcc crystal structure of a metal substrate. However, ice nucleation generally happened at the clearance between the graphene nanosheets, as shown in Figure 4a and Figure S5, when $\varepsilon_{w\text{-}m}$ was higher than the critical value mentioned above, i.e., 1.0 kcal/mol. In this case, a more hydrophilic metal substrate, rather than the hexagonal structure on the basal plane of the graphene nanosheet, dominates the position that a stable ice nucleus would form initially [60].

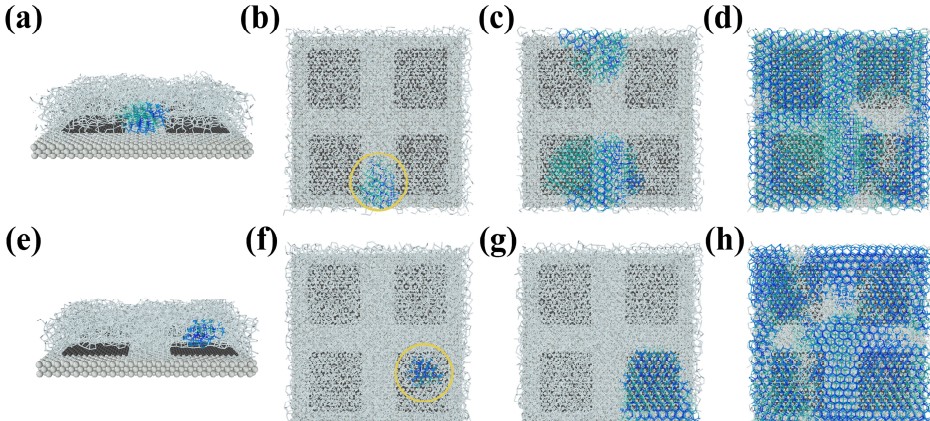

**Figure 4.** Snapshots of the preferential site of ice nucleation and subsequent growth of ice crystals on a metal–graphene surface. (**a**) Ice nucleation at the clearance of the surface, $\varepsilon_{w\text{-}m}$ = 4.0 kcal/mol. (**b**–**d**) Representative growth trajectories of an ice nucleus that appeared at the clearance, which is indicated by a yellow circle (from the top view). (**e**) Ice nucleation at the top of the graphene nanosheet, $\varepsilon_{w\text{-}m}$ = 1.0 kcal/mol. (**f**–**h**) Representative growth trajectories of an ice nucleus that appeared at the top of the graphene nanosheet. Silver and black spheres represent metal and carbon atoms. White sticks covered on them represent liquid water molecules, and blue and green sticks connect pairs of ice molecules with hexagonal and cubic structure orders, respectively.

Therefore, in this work, we draw the conclusion that an ice nucleus tends to form at the clearance of a surface with surface hydrophilicity discrepancy $D_\varepsilon$ larger than 0.87 kcal/mol. On the contrary, an ice nucleus tended to form at the top of the graphene nanosheets when the $D_\varepsilon$ was lower than 0.87 kcal/mol. Either at the clearance or at the top of the graphene nanosheets, an ice nucleus could grow continuously as the system temperature cooled down, as shown in Figure 4b–d or f–h. Nearly all liquid water molecules on the surface arranged in the regular order finally to form the ice structures. In addition, the growth rate of the ice crystal that nucleated initially at the clearance was much lower than that of the ice crystal that nucleated at the top of the graphene sheets, and the lowest growth rate of an ice crystal on the metal–graphene nanosheet surface could be found under the condition of the surface hydrophilicity discrepancy $D_\varepsilon$ being set to 7.87 kcal/mol.

### 3.3. Ice Growth and Stunting Effect of Boundary Misorientation

We also compared the mean growth rate of an ice nucleus under several surface conditions, including metal–graphene nanosheets, pure metal, pure graphene and graphene–graphene nanosheet systems, as shown in Figure 5. Different surface conditions and their models are illustrated in Figure 1b. The ice growth rate was calculated by the method described in Figure S6.

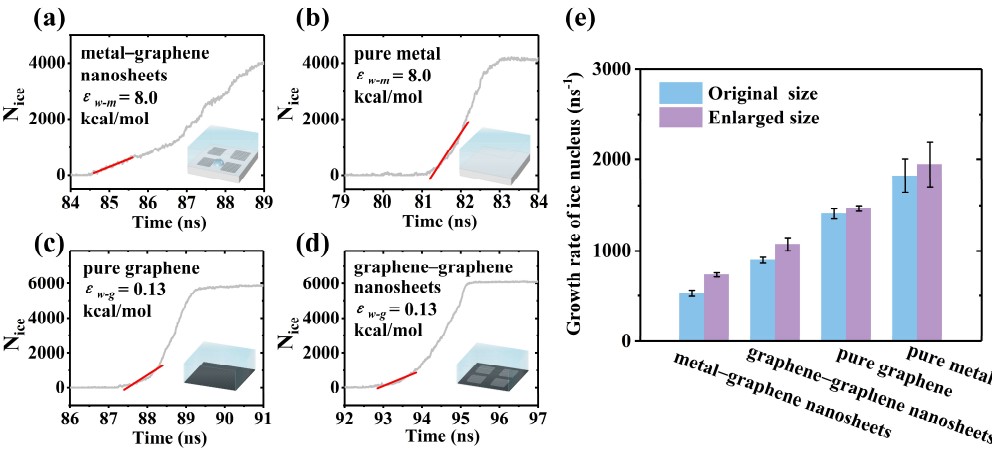

**Figure 5.** Growth rate of an ice nucleus on different surface configurations. (**a**) Number of water molecules in the ice nucleus, $N_{ice}$, changed with simulation time on the metal–graphene nanosheet surface. (**b**–**d**) $N_{ice}$ on a pure metal, pure graphene and graphene–graphene nanosheet surface, respectively. The data refer to $\varepsilon_{w-m}$ = 8.0 kcal/mol both in (**a**,**b**). The red line in each figure is the linear fitting of $N_{ice}$ within the first 1 ns after the initiation of icing. (**e**) Comparison of the ice growth rate on different surface configurations. Blue and purple bars represent the original and enlarged size of the simulation box.

The lowest growth rate of an ice crystal was on the metal–graphene nanosheet surface when $\varepsilon_{w-m}$ was set to 8.0 kcal/mol, i.e., 7.87 kcal/mol surface hydrophilicity discrepancy for the clearance configuration. Compared to the pure metal surface, this ice growth rate was decreased by 71.08%, as shown in Figure 5e. Similarly, for the graphene–graphene nanosheet surface, the growth rate of the ice nucleus decreased by 36.07% compared to that on the pure graphene surface. Ice growth was suppressed on both the metal and graphene substrate, while graphene nanosheets were anchored on them. These alternately distributed graphene nanosheets can restrict the growth of an ice nucleus in the direction parallel to the substrate [42]. An ice nucleus at the clearance tends to grow along the z axis, leading to the curved surface of the ice nucleus. According to the Gibbs–Thomson effect [43,62], the surface curvature of ice would cause additional pressure on the ice–water interface, resulting in an external resistance for the further growth of the ice nucleus. [42,63]. Hence, the free energy barrier for ice growth increases, and the freezing point of liquid water is decreased. However, the suppression effect was more prominent on the metal–graphene nanosheet surface configuration, owing to the proper surface hydrophilicity discrepancy and the rearranged interfacial water layer existing at the clearance configuration. Additionally, we found that ice nucleation occurred at a random position of the graphene–graphene nanosheet surface. The ice growth was suppressed only when the ice nucleus was formed at the clearance; otherwise, the ice nucleus grew on the graphene nanosheet without the suppression effect of the clearance. These results indicate that the suppression effect of the clearance generated by graphene nanosheets does occur both on metal and graphene substrates, and the optimal freezing delay ability of the surface can be achieved in the metal–graphene nanosheet surface system with a proper surface hydrophilicity discrepancy. The suppression effect of these surfaces followed the sequence of metal–graphene nanosheets > graphene–graphene nanosheets > pure graphene, as illustrated in Figure 5e.

On the other hand, we also constructed models with an enlarged size of the simulation box, which means an enlarged clearance width on these surfaces, and performed simulations to investigate the impact of clearance width on the growth of the ice nucleus. Details of the models are shown in Table S2. The simulation parameters of the enlarged systems were the same to that of the systems with the original size for each surface configuration. The initiation of ice nucleation on the metal–graphene nanosheet surface was greatly advanced (10–20 ns) by merely a 1 nm increase of the clearance width, and the ice growth rate increased noticeably by 40.02%, as shown in Figure S7 and Figure 5e. In the enlarged metal–graphene nanosheet system, the ice embryo could still nucleate at the enlarged clearance configuration. However, a stable ice nucleus formed before it contacted the lateral graphene nanosheets. Ice growth would not be restricted and the curved surface of an ice nucleus would not form. Consequently, the suppression effect of the clearance configuration disappeared, when the clearance width was larger than 2 nm on the surface both in metal–graphene nanosheet and graphene–graphene nanosheet systems. Therefore, our investigation leads to the conclusion that a pronounced ice inhibition will exhibit on the metal–graphene nanosheet surface configuration only when the clearance between graphene nanosheets is narrow enough. Further research needs to be developed for a critical size of the clearance configuration. In contrast, the enlarged size of the simulation box had no obvious influence on ice growth on the pure metal and pure graphene surfaces. For the graphene–graphene nanosheet surface, ice nucleation occurred at a random position on the surface in both the original and enlarged systems. When the clearance width was enlarged, ice growth would not be restricted by the graphene nanosheets, even if the ice nucleus was formed at the clearance. The growth rate of the ice nucleus on the graphene–graphene nanosheet surface slightly increased by 18.46% due to the enlarged clearance width, as shown in Figure 5e.

Figure 6a suggests that a stunting effect exists during the growth process of an ice crystal on a metal–graphene nanosheet surface. This is due to the boundary misorientation between ice crystals [47], as illustrated in Figure 6b. Obviously, the ice cluster marked with a red circle has a different orientation with the ice cluster on the right side. Generally, two growing ice nuclei contact each other, and the boundary misorientation forms. With the decreasing entropy and potential energy of a system, ice clusters with a boundary misorientation tended to transform into a consistent crystal, and the disorder boundary structure disappeared gradually, which suspended the growth of the ice crystal on the surface about 0.5–1.0 ns. This stunting effect could take a lot of time during the rapid growth of the ice crystal, as shown in Figure 6a. We also found that the duration of the stunting effect extended by increasing the contact area between ice clusters with different orientations, finally slowing down the growth rate of the ice crystal.

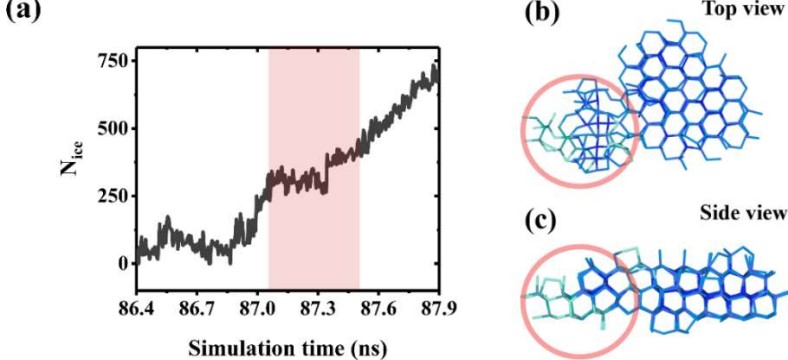

**Figure 6.** Stunting effect of a boundary misorientation during ice growth. (**a**) The inset shows the stunting period (shaded red) in the ice growth process caused by boundary misorientation between ice crystals. (**b**) Top view snapshots of boundary misorientation between ice crystals. (**c**) Side view of the boundary misorientation between ice crystals.

## 4. Conclusions

In summary, we carried out molecular dynamics simulations to investigate the freezing delay capability of metal–graphene nanosheet surfaces and gained a microscopic understanding of ice nucleation and growth, which were restricted at the clearance configuration. The interfacial water layer would form when the surface hydrophilicity of the metal substrate $\varepsilon_{w\text{-}m}$ was higher than 1.0 kcal/mol, separating other free water molecules from interacting with the metal substrate. This hindering effect of interfacial water layer competed with the increasing interaction strength between water molecules and the metal substrate. Ice nucleation was delayed when $\varepsilon_{w\text{-}m}$ ranged from 4.0 to 14.0 kcal/mol and the delayed time peaks to 5.25 ns with the surface hydrophilicity discrepancy $D_\varepsilon$ of 5.87 kcal/mol.

The nucleation position of an ice embryo changed from the top of graphene nanosheets to the clearance since the surface hydrophilicity of the metal substrate $\varepsilon_{w\text{-}m}$ surpassed 1.0 kcal/mol. The growth of the ice nucleus at the clearance was restricted by surrounding graphene nanosheets both on metal and graphene substrates. The ice nucleus grew with the lowest rate when $\varepsilon_{w\text{-}m}$ was 8.0 kcal/mol. An enlarged width of the clearance weakened the suppression effect remarkably, which is an important element in the design of anti-icing nanomaterials. Furthermore, the boundary misorientation between ice crystals can also suppress ice growth because of the stunting effect, which is proportional to the contact area between ice crystals. We believe these findings provide a simple model to explore the mechanism of nano-sized graphene and its derivatives on ice nucleation and growth, as well as their potential application for anti-icing materials.

**Supplementary Materials:** The following supporting information can be downloaded at: https://www.mdpi.com/article/10.3390/coatings12010052/s1, Figure S1: Morphology of the simulation system. (a) A typical metal–graphene nanosheets surface, containing four single graphene nanosheets anchored alternately on a metal substrate. (b) Entire simulation box with a water slab covered on the surface. Atoms of metal substrate and graphene nanosheets are colored by silver and black respectively, hydrogen bonds between liquid water molecules are represented by white sticks with a cut off distance of 3.2 Å, which is realized by the Create Bonds modifier in OVITO software; Figure S2: Example of a simulation box used in ice freezing. Periodic boundary conditions (PBC) were applied in x, y and z directions. A 7-nm-thick void region was incorporated in the simulation box to avoid the undesired effect of metal substrate in the adjacent simulation box caused by the PBC in the z direction; Figure S3: Ice formation on the metal–graphene nanosheets surface. (a) Stacking disordered ice formed on the surface ($\varepsilon_{w\text{-}m}$ = 1.0 kcal/mol), from a cross section view. Random layers of ice $I_h$ and ice $I_c$ were colored in bule and green respectively. (b) Count of water molecules above the surface along z direction, which depicts the regular order of ice; Figure S4: Ice growth at the top the graphene nanosheets with a surface hydrophilicity of metal substrate of 0.13 kcal/mol, which is equal to the surface hydrophilicity of graphene $\varepsilon_{w\text{-}g}$. (a) Ice embryo nucleating at the top the graphene nanosheets owing to the repeated hexagonal carbon ring structure on graphene. (b–d) Growth of the ice crystal on the graphene nanosheets; Figure S5: Ice growth at the clearance, between the peripheral graphene nanosheets, with a surface hydrophilicity of metal substrate of 4.0 kcal/mol. (a) Ice embryo nucleated at the clearance configuration. (b–d) Restricted growth of the ice crystal on the metal–graphene nanosheets surface; Figure S6: Number of water molecules in ice nucleus changed with simulation time on metal–graphene nanosheets surface with different surface hydrophilicity of metal substrate; Figure S7: Initiation time of icing on metal–graphene nanosheets with different size of simulation box. The clearance width is about 2.0 nm in original sized system, whereas it is 3.0 nm in enlarged sized system; Table S1: The size of simulation box; Table S2: The interaction parameters for the Model Systems in this work.

**Author Contributions:** Conceptualization, Y.S.; methodology, Y.X.; validation, B.J., Y.S., J.T., Y.X., H.C., and S.L.; writing—original draft preparation, B.J. and Y.S.; writing—review and editing, J.T. and Y.S.; visualization, W.L. and X.X.; funding acquisition, J.T. and Y.S. All authors have read and agreed to the published version of the manuscript.

**Funding:** This work was supported by the National Natural Science Foundation of China (Nos. 52075246, 12002364, and U1937206), the Natural Science Foundation of Jiangsu Province (No. BK20211568), the Project Funded by China Postdoctoral Science Foundation (No. 2019M661826), the

Open Fund of Key Laboratory of Icing and Anti/De-icing (Nos. IADL20190202, IADL20200407) and the NUAA Innovation Program for Graduate Education (kfjj20200613, kfjj20200605).

**Institutional Review Board Statement:** Not applicable.

**Informed Consent Statement:** Not applicable.

**Data Availability Statement:** The data presented in this study are available within the article.

**Conflicts of Interest:** The authors declare no conflict of interest.

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
