# Peer review of "Patterning Configuration of Surface Hydrophilicity by Graphene Nanosheet towards the Inhibition of Ice Nucleation and Growth"

_coatings, doi:10.3390/coatings12010052_

Round 1

Reviewer 1 Report

The manuscript is poorly written, unclear, and poorly organized. The discussion and conclusions are not supported by the modeling data. The important information on the details of the method used and the way of data interpretation are either not given or transmitted to the Supporting information. It is unclear, how the data, presented in figures (for example Fig.6) were obtained and interpreted.

  1. What does mean the phrase: “Lz was set to be sufficiently large to eliminate the IMAGE EFFECT in the z direction.“?
  2. The coarse-grained mW water model used in this study should be described and its utilization for the purposes of this study should be substantiated.
  3. The phrase “After 0.2 ps of relaxation at temperature 290 K, the whole system was cooled down to 200 K with a cooling rate of 0.9 K/ns in the NVT ensemble.” Please, indicate explicitly that the freezing delay was studied at 200 K. If so, why do authors consider the heterogeneous nucleation only and neglect the homogeneous one (DOI: 10.1146/annurev.pc.34.100183.003113).
  4. The claim “the formed interfacial water layer acts as a barrier and hinders the nucleation of ice nucleus” requires explanation. How authors have detected such a layer and how they substantiated that this layer hinders the nucleation?
  5. In Fig. 4, the time intervals for linear fitting were chosen arbitrarily without justification and agreement with the presented data.
  6. The claim: “These alternately distributed graphene nanosheets can restrict the growth of ice nucleus at the clearance, leading to the formation of a curved surface of ice crystal. As a result, the growth of the ice nucleus is suppressed owing to the Gibbs-Thompson effect in both surface conditions” is given without any explanation and justification. It is unclear, how the Gibbs-Thompson effect can SUPPRESS the nucleation.
  7. It is written in the manuscript:” To our surprise, the initiation of ice nucleation on metal-graphene nanosheets surface is greatly advanced (10-20 ns) by merely 1 nm increase of the clearance width, and the ice growth rate increased noticeably by 40.02%, as shown in Figure 4(b). “ However, data presented in Figure 4 do not allow to make such a conclusion.
  8. The claim: “… the nucleus will form at the clearance owing to larger contact area between ice nucleus and graphene, i.e., lower free energy barrier for ice nucleation” seems incorrect and was given without any explanation or discussion.
  9. The same situation with the claim: “The ice cubicity is mainly influenced by the interfacial water layer, which regulates the initiation time of ice nucleation on the metal-graphene nanosheets surface. Higher interaction strength between metal substrate and water molecule results in orderly rearrangement of water molecules, which is convenient for the formation of ice Ih rather than ice Ic.” Explanation and discussion based on the obtained results are required here.

Author Response

Response to Reviewer

Thank you very much for your careful reading, helpful comments, and constructive suggestions, which has significantly improved the presentation of our work. We have carefully considered the comments and revised our manuscript accordingly. Also, the supplements were clearly highlighted using modifing mode in revised manuscript. The main corrections and the response to reviewer’s comments are listed as follows:

We look forward to hearing from you soon.

Best regards!

Prof. Jie Tao and Assoc. Prof. Yizhou Shen

Nanjing University of Aeronautics and Astronautics, P. R. China

The manuscript is poorly written, unclear, and poorly organized. The discussion and conclusions are not supported by the modeling data. The important information on the details of the method used and the way of data interpretation are either not given or transmitted to the Supporting information. It is unclear, how the data, presented in figures (for example Fig.6) were obtained and interpreted.

  1. What does mean the phrase: “Lz was set to be sufficiently large to eliminate the IMAGE EFFECT in the z direction.”?

Author reply: We are very sorry for our unclear description, and set the periodic boundary condition (PBC) in our simulation. Also, a void region was set to avoid the effect of metal substrate in the adjacent simulation box (image), as shown in Figure R1. The phrase has been rewrote in the revised manuscript.

Figure R1. The simulation box and the void region in it.

  1. The coarse-grained mW water model used in this study should be described and its utilization for the purposes of this study should be substantiated.

Author reply: Thank you for the above suggestions. We added the description about the coarse-grained mW water model, and explained the purpose we utilized this model for simulation in section 2 of System and simulations. In addition, the detailed information of the mW model was added to the Supplementary Note 1.

This paper mainly focuses on the inhibition of ice nucleation and growth on the hydrophobicity-designed surface, while ignoring hydrogen bond and electrostatic forces between water molecules (DOI: 10.1038/ncomms15372). The mW is a suitable water model for the work of this paper since it is monatomic. In addition, the mW water model can also improve the computational efficiency of simulations comparing with full atomic water models.

  1. The phrase “After 0.2 ps of relaxation at temperature 290 K, the whole system was cooled down to 200 K with a cooling rate of 0.9 K/ns in the NVT ensemble.” Please, indicate explicitly that the freezing delay was studied at 200 K. If so, why do authors consider the heterogeneous nucleation only and neglect the homogeneous one (DOI: 10.1146/annurev.pc.34.100183.003113).

Author reply: Thank you for pointing out this problem in manuscript. The system was cooled from 290 K to 200 K gradually, and we studied the process of water freezing during this quenching period. The temperature of homogeneous ice nucleation for the mW water model is Thomo f= 201 ± 1 K (DOI: 10.1021/ja411507a). However, in each simulation of our work, the temperature of ice nucleation was higher than Thomo f. Meantime, in the results of our simulations, ice nucleation always happened on the surface (substrate or graphene), but not in the bulk of liquid water. No homogeneous ice nucleation was found in our simulations, so we only considered the heterogeneous ice nucleation in this work.

  1. The claim “the formed interfacial water layer acts as a barrier and hinders the nucleation of ice nucleus” requires explanation. How authors have detected such a layer and how they substantiated that this layer hinders the nucleation?

Author reply: In revised version, we added snapshots of the interfacial water layer in the revised manuscript (see Figure 3) to discuss the effect of this layer on nucleation. The interfacial water layer could be detected by the number of water molecules along the z direction, and a dense layer of water molecules was formed on metal substrate.

When εw-m increased to 1.0 kcal/mol, the interfacial water layer was formed on the substrate. Thereafter, the initiation time of icing was delayed, as shown in Figure 2(b). Therefore, we considered that the interfacial water layer was the reason for the delayed initiation time of icing. Water molecules in this layer was close packed in fcc structure that differs from the structure of ice crystal, which was not beneficial for ice nucleation. Meantime, other free water molecules were separated from the metal substrate by the interfacial water layer, which decreased the interaction between free water molecule and the metal substrate. Therefore, the interfacial water layer acts as a barrier and hinders the nucleation of ice nucleus.

  1. In Figure 4, the time intervals for linear fitting were chosen arbitrarily without justification and agreement with the presented data.

Author reply: We updated and corrected Figure 4 ( i.e. the growth curve of ice nucleus on pure metal surface with εw-m of 8.0 kcal/mol), which was regulated to Figure 5 in revised manuscript. This study mainly focuses on the initial stage of ice growth, so the time interval for linear fitting is the first 1 ns after the initiation of icing.

  1. The claim: “These alternately distributed graphene nanosheets can restrict the growth of ice nucleus at the clearance, leading to the formation of a curved surface of ice crystal. As a result, the growth of the ice nucleus is suppressed owing to the Gibbs-Thompson effect in both surface conditions” is given without any explanation and justification. It is unclear, how the Gibbs-Thompson effect can SUPPRESS the nucleation.

Author reply: Thank you for pointing out this problem in manuscript. We added some contents to describe the Gibbs-Thompson effect on ice nucleation in the revised manuscript.

According to the Gibbs-Thompson effect, the surface curvature of ice would cause additional pressure on ice-water interface, resulting in an external resistance for the further growth of ice nucleus. Hence, the free energy barrier for ice growth increases, and the freezing point of liquid water is decreased. The similar description was reported in the following literatures (DOI: 10.1016/j.scriptamat.2004.12.026), ( DOI: 10.1039/b809438c).

  1. It is written in the manuscript:" To our surprise, the initiation of ice nucleation on metal-graphene nanosheets surface is greatly advanced (10-20 ns) by merely 1 nm increase of the clearance width, and the ice growth rate increased noticeably by 40.02%, as shown in Figure 4(b). " However, data presented in Figure 4 do not allow to make such a conclusion.

Author reply: We are very sorry for our negligence, Figure S6 has been added in the Supplementary file for comparing the initiation time of icing in systems with different sizes, and the increase of ice growth rate is shown in Figure 5(e). Thank you so much for your helpful comments.

  1. The claim: “… the nucleus will form at the clearance owing to larger contact area between ice nucleus and graphene, i.e., lower free energy barrier for ice nucleation” seems incorrect and was given without any explanation or discussion.

Author reply: We checked the results of our simulation and rewrote this part in the revised manuscript.

In both original and enlarged systems, ice nucleation occurs at random position of the graphene-graphene nanosheets surface. The absence of surface hydrophilicity discrepancy and interfacial water layer are the main reasons for the random position of ice nucleation. Ice growth is suppressed only when the ice nucleus is formed at the clearance in original system. When the clearance width is enlarged, the growth of ice nucleus would not be restricted by the graphene nanosheets, even if the ice nucleus was formed at the clearance.

  1. The same situation with the claim: “The ice cubicity is mainly influenced by the interfacial water layer, which regulates the initiation time of ice nucleation on the metal-graphene nanosheets surface. Higher interaction strength between metal substrate and water molecule results in orderly rearrangement of water molecules, which is convenient for the formation of ice Ih rather than ice Ic.” Explanation and discussion based on the obtained results are required here.

Author reply: Thank you so much for your constructive suggestion.  

Our discussion about the influence factors of ice cubicity is an inference at present, and we plan to further prove it in the future work. Meantime, this part of discussion is not closely related to the content of this paper (the suppression effect of the clearance on ice freezing), so we want to delete the part of discussion about the ice cubicity.

Using the identified diamond structure modifier in OVITO software (DOI: 10.1016/j.cpc.2016.04.001), we obtained the number of water molecules belong to cubic and hexagonal ice, Nc and Nh. After the ending of the freezing process, the value of ice cubicity C was calculated by formula : C = Nc / Nice , where Nice is the total number of water molecules in ice cluster.

Reviewer 2 Report

The subject is interesting and it is worthy of publication. However, the following issues may be further considered for improve the MS.

My comments.

Isn't the title too general? Perhaps the title should contain information about graphene nanosheet?
Resolution of figures should be higher.

Author Response

Response to Reviewer

Thank you very much for your constructive comments concerning our manuscript. We have revised carefully the paper and addressed all these comments. Also, the modifications and supplements were clearly highlighted using modification mode in revised manuscript. The main corrections and the response to reviewer’s comments are listed as follows:

We look forward to hearing from you soon.

Best regards!

Prof. Jie Tao and Assoc. Prof. Yizhou Shen

Nanjing University of Aeronautics and Astronautics, P. R. China

The subject is interesting and it is worthy of publication. However, the following issues may be further considered for improve the MS.

  1. Isn't the title too general? Perhaps the title should contain information about graphene nanosheet?

Author reply: Many thanks! We have changed the title to ‘Patterning Configuration of Surface Hydrophilicity by Graphene Nanosheet towards the Inhibition of Ice Nucleation and Growth’.

  1. Resolution of figures should be higher.

Author reply: Thanks for your kind suggestion. We have improved the resolution of Figures in the revised manuscript.

Reviewer 3 Report

The manuscript reports a molecular dynamics based investigation of the formation and growth of ice on metal-graphene frameworks. The study is very interesting, and is of relevance to the understanding of ice nucleation and its modification with engineered surfaces. Hence, I recommend that the study could be accepted for publication in Coatings. However, a few minor issues need to be sorted out (as listed below) in a revised manuscript before it can be accepted for publication.

Comments:

  1. Lines #27-29: Do the authors mean that whether hexagonal or cubic ice crystals are formed is dependent on the time at which ice nucleation is triggered? Please clarify this statement.
  2. Line #86: What is the criterion for Lz in order to avoid the 'image effect'? In other words, what is the least value of Lz that would eliminate 'image effect'?
  3. Line #88: What does this distance of 6 Å (0.6 nm) refer to? Please mark this clearly in the figure.
  4. Please be more specific when referring to the supplementary material throughout the manuscript. Please provide the Table, Fig., or Sec, number being referred to (for example, in line #98).
  5. What is the sensitivity of the observed phenomena (of ice nucleation) to the chosen model of water-substrate interaction potential? In other words, can it be scientifically shown or argued that choosing a different inter-atomic potential (other than L-J) would result in similar observations as reported here?
  6. It is not clear how periodic boundary condition (PBC) is applied in the direction normal to the surface pointing towards the water side of the domain. Please clarify this (perhaps, in the supplementary material). 
  7. Why is there an increase in ice nucleation initiation time when the interaction strength is increased from 0.13 kcal/mol to 0.5 kcal/mol in Fig. 2a?
  8. The vertical axis in Fig. 2b is marked as 'Simulation time' -- is this the same as the initiation time of icing? Similarly, the vertical axis in Fig. 2a ('number of water molecules') is different from what is written in the caption ('efficiency'), Please check, correct, and be consistent in the terms used (also, please check line #132 where the term 'freezing efficiency' is used without a proper definition).
  9. Line #152: Was εw-m fixed or varied?
  10. It is not clear what Ih and Ic refer to in Fig. S2a, Please mark these in an enlarged manner in the figure. Secondly, the vertical axis in Fig. S2b says 'Count' whereas the caption refers to 'density'. Please clarify how these are related, and please be consistent in the usage of terms throughout the manuscript.

Author Response

Response to Reviewer

Thank you very much for your careful reading, helpful comments, and constructive suggestions concerning our manuscript. We have revised carefully the paper and addressed all these comments. Also, the supplements were clearly highlighted using modification mode in revised manuscript. The main corrections and the response to reviewer’s comments are listed as follows:

We look forward to hearing from you soon.

Best regards!

Prof. Jie Tao and Assoc. Prof. Yizhou Shen

Nanjing University of Aeronautics and Astronautics, P. R. China

The manuscript reports a molecular dynamics based investigation of the formation and growth of ice on metal-graphene frameworks. The study is very interesting, and is of relevance to the understanding of ice nucleation and its modification with engineered surfaces. Hence, I recommend that the study could be accepted for publication in Coatings. However, a few minor issues need to be sorted out (as listed below) in a revised manuscript before it can be accepted for publication.

  1. Lines #27-29: Do the authors mean that whether hexagonal or cubic ice crystals are formed is dependent on the time at which ice nucleation is triggered? Please clarify this statement.

Author reply: We are very sorry for our unclear statement. Whether hexagonal or cubic ice are formed is dependent on the structure of the interfacial water layer. We have made corrections in the revised manuscript.

Since our discussion about the influence factors of ice cubicity is an inference at present, we plan to further prove it in the future work. Meantime, this part of discussion is not closely related to the content of this paper (the suppression effect of the clearance on ice freezing), so we delete the part of discussion about the ice cubicity.

  1. Line #86: What is the criterion for Lz in order to avoid the 'image effect'? In other words, what is the least value of Lz that would eliminate 'image effect'?

Author reply: The undesired image effect caused by periodic boundary conditions (PBC) was mentioned in some researches, and 3-nm-thick void region in simulation box would be enough to avoid this effect (the motion of water molecules would not be disturbed by the substate in the adjacent simulation box in z direction). (DIO: 10.1021/acs.jpcc.8b07779). We considered that the thickness of the void region should be at least 3 nm in our work. The void region in our simulation box is about 7-nm-thick, so we believe the influence caused by the PBC has been eliminated. The description has been rewritten in the revised manuscript.

  1. Line #88: What does this distance of 6 Å (0.6 nm) refer to? Please mark this clearly in the figure.

Author reply: Thank you for the above suggestion. The distance of 6 Å is the vertical distance between the graphene sheet and the substrate in the direction normal to the surface, and we have marked it as c in Figure 1(a).

  1. Please be more specific when referring to the supplementary material throughout the manuscript. Please provide the Table, Fig., or Sec, number being referred to (for example, in line #98).

Author reply: The specific Table and Figure numbers were added in the new revised manuscript (Line#110 and #121). Thank you for your kindly reminding.

  1. What is the sensitivity of the observed phenomena (of ice nucleation) to the chosen model of water-substrate interaction potential? In other words, can it be scientifically shown or argued that choosing a different inter-atomic potential (other than L-J) would result in similar observations as reported here?

Author reply: The L-J potential is a suitable and common potential for water-substrate interaction, and it has been widely used in the other researches (DOI: 10.1021/jacs.5b08748 and DOI: 10.1039/c6cp04382h).

Two-body term of mW model could also be used to describe water-substrate interaction (DOI: 10.1021/acs.jpcc.5b09740), which was given by Stillinger-Weber (SW) potential. Similarly, heterogeneous ice nucleation on surface was observed in this literature.

  1. It is not clear how periodic boundary condition (PBC) is applied in the direction normal to the surface pointing towards the water side of the domain. Please clarify this (perhaps, in the supplementary material).

Author reply: Periodic boundary conditions (PBC) were applied in x, y and z directions, so that particles interact across the boundary, and they can exit one end of the box and re-enter the other end. A 7-nm-thick void region was incorporated in the simulation box to avoid the undesired effect of metal substrate in the adjacent simulation box caused by the PBC in the z direction.

We added a Figure S2 in the supplementary file to explain how PBC was applied.

  1. Why is there an increase in ice nucleation initiation time when the interaction strength is increased from 0.13 kcal/mol to 0.5 kcal/mol in Fig. 2a?

Author reply: The interfacial water layer is formed gradually with the increasing εw-m. When the εw-m is increased to 0.5 kcal/mol, the interfacial water layer begins to form on the substate, as shown in Figure R1. Water molecules in this layer (colored in red) is close packed in fcc structure that differs from the structure of ice crystal, which is not beneficial for ice nucleation. On the other hand, other free water molecules are separated from the metal substrate by the interfacial water layer. This incomplete water layer can partly hinder the water-substrate interaction, and the ice nucleation initiation time is delayed. Therefore, the initiation time of icing rises when εw-m is increased from 0.13 kcal/mol to 0.5 kcal/mol. However, with the increasing of εw-m, the initiation time of ice nucleation is advanced, as shown in Figure 2 in revised manuscript.

Figure R1. The formation of the interfacial water layer on the substrate.

  1. The vertical axis in Fig. 2b is marked as 'Simulation time' -- is this the same as the initiation time of icing? Similarly, the vertical axis in Fig. 2a ('number of water molecules') is different from what is written in the caption ('efficiency'), Please check, correct, and be consistent in the terms used (also, please check line #132 where the term 'freezing efficiency' is used without a proper definition).

Author reply: Thank you for your kindly reminding. We have corrected the terms on the axes and in the captions according to your comments.

  1. Line #152: Was εw-m fixed or varied?.

Author reply: We are very sorry for our unclear description. εw-m was fixed to 1.0 kcal/mol for the inset of Figure 2(b). We have corrected it in the caption of Figure 2 in revised manuscript.

  1. It is not clear what Ih and Ic refer to in Fig. S2a, Please mark these in an enlarged manner in the figure. Secondly, the vertical axis in Fig. S2b says 'Count' whereas the caption refers to 'density'. Please clarify how these are related, and please be consistent in the usage of terms throughout the manuscript.

Author reply: Thank you for the above suggestion. We marked Ih and Ic in an enlarged manner in Figure S3(a) and corrected the caption of Figure S3(b).

Reviewer 4 Report

Effect of Graphene and modified graphene on icing is a very interesting phenomenon and both experimental and theoretical/numerical simulations are needed to understand the icing formation and eventual elimination of icing. In this respect, this paper uses molecular simulations to understand the surface effects of graphene and metal-graphene films on ice formation and the authors show that nucleation and the first frozen interface can form independent of the bulk above and this is related to the hydrophilicity of the graphene or modified graphene layer. In general, I found the manuscript interesting and eventually suitable for the journal Coatings after the authors make following amendments to their manuscript:

  1. Please improve the introduction section some more by discussing and including a number of references that have been overlooked. Please involve them in the introduction: "Mechanisms of Surface Icing and Deicing Technologies." Ice Adhesion: Mechanism, Measurement and Mitigation(2020): 325-359 (https://doi.org/10.1002/9781119640523.ch11 ). Please also mention and discuss importance of graphene as potential “heater” to prevent icing like joule heating or electrical heating referring to these works: Composites Science and Technology 164 (2018): 65-73; RSC advances30 (2018): 16815-16823; Composites Part B: Engineering 162 (2019): 600-610. 

Author Response

Response to Reviewer

Thank you very much for your helpful and constructive comments concerning our manuscript. We have revised carefully the paper and addressed all these comments. Also, the supplements were clearly highlighted using purple in revised manuscript. The main corrections and the response to reviewer’s comments are listed as follows:

We look forward to hearing from you soon.

Best regards!

Prof. Jie Tao and Assoc. Prof. Yizhou Shen

Nanjing University of Aeronautics and Astronautics, P. R. China

Effect of Graphene and modified graphene on icing is a very interesting phenomenon and both experimental and theoretical/numerical simulations are needed to understand the icing formation and eventual elimination of icing. In this respect, this paper uses molecular simulations to understand the surface effects of graphene and metal-graphene films on ice formation and the authors show that nucleation and the first frozen interface can form independent of the bulk above and this is related to the hydrophilicity of the graphene or modified graphene layer. In general, I found the manuscript interesting and eventually suitable for the journal Coatings after the authors make following amendments to their manuscript:

  1. Please improve the introduction section some more by discussing and including a number of references that have been overlooked. Please involve them in the introduction: "Mechanisms of Surface Icing and Deicing Technologies." Ice Adhesion: Mechanism, Measurement and Mitigation(2020): 325-359 (https://doi.org/10.1002/9781119640523.ch11). Please also mention and discuss importance of graphene as potential “heater” to prevent icing like joule heating or electrical heating referring to these works: Composites Science and Technology 164 (2018): 65-73; RSC advances30 (2018): 16815-16823; Composites Part B: Engineering 162 (2019): 600-610. 

Author reply: Thanks very much! According to your suggestion, we have discussed and added several references in the introduction section, and the importance of graphene material as potential “heater” to prevent ice freezing has been discussed. Specific references are listed as follows:

  1. Lin, Y.; Chen, H.; Wang, G. Liu, A. Recent Progress in Preparation and Anti-Icing Applications of Superhydrophobic Coatings. Coatings, 2018, 8.
  2. Yue, X.; Liu, W. Wang, Y. Effects of black silicon surface structures on wetting behaviors, single water droplet icing and frosting under natural convection conditions. Surface and Coatings Technology, 2016, 307, 278-286.
  3. Wang, Y. Wang, Z.-g. Sessile droplet freezing on polished and micro-micro-hierarchical silicon surfaces. Applied Thermal Engineering, 2018, 137, 66-73.
  4. He, Z.Y.; Xie, W.J.; Liu, Z.Q.; Liu, G.M.; Wang, Z.W.; Gao, Y.Q. Wang, J.J. Tuning ice nucleation with counterions on polyelectrolyte brush surfaces. Science Advances, 2016, 2.
  5. Redondo, O.; Prolongo, S.G.; Campo, M.; Sbarufatti, C. Giglio, M. Anti-icing and de-icing coatings based Joule's heating of graphene nanoplatelets. Composites Science and Technology, 2018, 164, 65-73.
  6. Vertuccio, L.; De Santis, F.; Pantani, R.; Lafdi, K. Guadagno, L. Effective de-icing skin using graphene-based flexible heater. Composites Part B: Engineering, 2019, 162, 600-610.
  7. Karim, N.; Zhang, M.; Afroj, S.; Koncherry, V.; Potluri, P. Novoselov, K.S. Graphene-based surface heater for de-icing applications. RSC Advances, 2018, 8, 16815-16823.
  8. Chen, L.; Zhang, Y. Wu, Q. Effect of Graphene Coating on the Heat Transfer Performance of a Composite Anti-/Deicing Component. Coatings, 2017, 7.
  9. Zhang, X.-X. Chen, M. Icephobicity of Functionalized Graphene Surfaces. Journal of Nanomaterials, 2016, 2016, 1-8.
  10. Akhtar, N.; Anemone, G.; Farias, D. Holst, B. Fluorinated graphene provides long lasting ice inhibition in high humidity. Carbon, 2019, 141, 451-456.

Round 2

Reviewer 1 Report

After revision, the manuscript remained unclear, poorly written, and contains a lot of erroneous results. 

From the description of mW  water model added to the manuscript, it can be concluded that the model is not appropriate to describe the ice/substrate interactions, because this model completely ignores the main types of interaction (hydrogen bonding and electrostatic) responsible for the ice/substrate adhesion (DOI: doi.org/10.1021/jp9632145, 10.1021/acs.jpcc.8b12435, 10.3390/coatings10070648).

It is still unclear why do authors consider the heterogeneous nucleation only and neglect the homogeneous one. Their reply is based on the erroneous statement that “the temperature of homogeneous ice nucleation for the mW water model is Thomo f= 201 ± 1 K “. The nucleation is a stochastic process and can take at any instant for temperatures less than the water triple point. Thus, if mW water model predicts the only temperature of homogeneous nucleation, this means that the model is invalid. As an example, the Reviewer would like to remind that the limiting temperature of bulk water supercooling is -42 deg., however, this fact does not result in the homogeneous crystallization in any bulk water vessel at T=-42 deg., and a lot of freezing events will take place at a temperature around, say, 0 deg.

This list of manuscript flaws can be continued further (see the first Review). However, the two points mentioned above are enough to conclude that this manuscript cannot be recommended for publication.

Author Response

Response to Reviewer

Thank you very much for your careful reading, helpful comments, and constructive suggestions, which has significantly improved the presentation of our manuscript. We have carefully considered the comments and revised our manuscript accordingly. Also, the modifications and supplements were clearly highlighted using modifying mode in revised manuscript. The main corrections and the response to reviewer’s comments are listed as follows:

We look forward to hearing from you soon.

Best regards!

Prof. Jie Tao and Assoc. Prof. Yizhou Shen

Nanjing University of Aeronautics and Astronautics, P. R. China

  1. From the description of mW water model added to the manuscript, it can be concluded that the model is not appropriate to describe the ice/substrate interactions, because this model completely ignores the main types of interaction (hydrogen bonding and electrostatic) responsible for the ice/substrate adhesion (DOI: doi.org/10.1021/jp9632145,10.1021/acs.jpcc.8b12435,10.3390/coatings10070648).

Author reply: We are very sorry for our unclear description. It is true that all-atoms water model is more accurate than coarse-grained water model in some simulation systems (such as system with external ions, system with the external electric field, system considering ice adhesion). However, this paper mainly focuses on the phase transition of liquid water, as well as the structure and size of the ice nucleus. The coarse-grained mW water model represents a water molecule as a single particle with short-range anisotropic interactions that mimic hydrogen bonds. The mW model can reproduce the structure and phase behavior of water with accuracy comparable to all-atoms models (DOI: 10.1021/jp805227c), and it has been widely used to study the heterogeneous ice nucleation of water because of its satisfactory accuracy and high computational efficiency. (DOI: 10.1021/acs.jpcc.5b09740, 10.1021/acs.jpcc.5b09740, 10.1021/jacs.5b08748, 10.1016/j.apsusc.2020.145520). Therefore, we consider that the mW is a suitable water model for the work of this paper.

  1. It is still unclear why do authors consider the heterogeneous nucleation only and neglect the homogeneous one. Their reply is based on the erroneous statement that “the temperature of homogeneous ice nucleation for the mW water model is Thomo f = 201 ± 1 K ”. The nucleation is a stochastic process and can take at any instant for temperatures less than the water triple point. Thus, if mW water model predicts the only temperature of homogeneous nucleation, this means that the model is invalid. As an example, the Reviewer would like to remind that the limiting temperature of bulk water supercooling is -42 deg., however, this fact does not result in the homogeneous crystallization in any bulk water vessel at T=-42 deg., and a lot of freezing events will take place at a temperature around, say, 0 deg.

Author reply: Thank you for underlining this deficiency. It is really true as the Reviewer suggested that the nucleation is a stochastic process and can take place at any temperature less than the water triple point. However, the above conclusion is true in the constant temperature system (keep the system at any temperature lower than 273 K for a long time), and our simulation system is a quenching system. The phase transition process of water needs a long induction period, and the induction period is directly related to the system temperature, which means that nucleation at higher temperature requires a longer induction period ( >100 ns at 240 K, > 45 ns at 220 K, ~ 10 ns at 200 K).

We performed isothermal simulations of homogeneous nucleation at temperatures of 200, 220 and 240 K. The induction period of nucleation were measured from the potential energy of the system, as shown in Figure R1.

Figure R1. The total potential energy of homogeneous systems with different isothermal temperature. The potential energy will drop obviously when nucleation happens, so the beginning of ice nucleation can be represented by the time when the potential energy decreases. (a) 240 K. Nucleation did not happen within 100 ns. The induction period is longer than 100 ns. (b) 220 K. The induction period is longer than 45 ns. (c) 200 K. Homogeneous nucleation happened after nearly 10 ns.

With the cooling rate of 1 K/ns, noticeable homogeneous nucleation did not occur until the temperature dropped from 298 K to about 201 K for mW model (DOI:10.1021/ja411507a). The heterogeneous nucleation would take place preferentially before the homogeneous nucleation, owing to lower activation free energy barrier for heterogeneous nucleation on solid surfaces. The nucleation temperatures in the results of our cooling simulation were higher than 201 K, and no visible homogeneous ice nucleation was observed during the freezing process. In addition, liquid water in our system had been completely frozen within 1- 4 ns from the beginning of heterogeneous nucleation, and homogeneous nucleation would not happen in such a short time frame. Therefore, we consider that the heterogeneous nucleation is the main part in our simulation results.

Reviewer 4 Report

Please also add and discuss as mentioned in the earlier report. 

Ice Adhesion: Mechanism, Measurement and Mitigation(2020): 325-359
(https://doi.org/10.1002/9781119640523.ch11 ) 

Author Response

Response to Reviewer

Thank you very much for your helpful comments concerning our manuscript. We have revised carefully the paper and addressed the comments. Also, the supplements were clearly highlighted using modifying mode in revised manuscript. The main corrections and the response to reviewer’s comments are listed as follows:

We look forward to hearing from you soon.

Best regards!

Prof. Jie Tao and Assoc. Prof. Yizhou Shen

Nanjing University of Aeronautics and Astronautics, P. R. China

  1. Please also add and discuss as mentioned in the earlier report.

Ice Adhesion: Mechanism, Measurement and Mitigation(2020): 325-359

(https://doi.org/10.1002/9781119640523.ch11 )

Author reply: Thank you for your suggestion, we have discussed and added the reference in the revised manuscript.
